REGISTERED REPORT PROTOCOL

# A scoping review protocol on childhood immunization reminder strategies available to parents in Canada and the United States of America

Matilda Anim-Larbi[1]*, Vivian Puplampu[1], Sithokozile Maposa[1,2], Akram Mahani[3], Mary Chipanshi[4‡]

1 Faculty of Nursing, University of Regina, Regina, Saskatchewan, Canada, 2 College of Nursing, University of Saskatchewan, Prince Albert, Saskatchewan, Canada, 3 Johnson Shoyama Graduate School of Public Policy, University of Regina, Regina, Saskatchewan, Canada, 4 Dr. John Archer Library & Archives, University of Regina, Regina, Saskatchewan, Canada

☯ These authors contributed equally to this work.
‡ MC also contributed equally to this work.
* max001@uregina.ca

## Abstract

### Introduction

Vaccine-preventable diseases continue to cause morbidity and mortality despite the introduction of childhood immunizations. Recent media reports from Canada and the United States of America (USA) have highlighted a rise in childhood illnesses like measles, which could have been prevented with vaccines. Parents play a pivotal role in ensuring their children receive timely vaccinations. Immunization reminders can help parents who forget or miss vaccination appointments. In the USA, current literature indicates that Black children have lower vaccination rates than other racialized children and vaccine reminders may improve measles vaccine uptake among Black parents. However, there is limited data in Canada on vaccine uptake in children of Black parents, with evidence suggesting vaccine hesitancy among the Black population.

### Objective

This scoping review aims to map out existing literature on immunization reminder strategies among parents to identify their impact in improving childhood vaccination rates and promoting child health.

**Data availability statement:** The authors will keep the data that underlies the review in safety in a password-protected computer. Data from the review will be shared with the public using the University of Regina data-sharing space with the public, Dataverse.

**Funding:** The author(s) received no specific funding for this work.

**Competing interests:** The authors have declared that no competing interests exist.

## Inclusion criteria

The review will include studies conducted in Canada and the United States of America that focus on immunization reminders for parents who have children under six years and published in English between 2015 and 2025.

## Methods

Database and hand-searching of journals and gray literature will be carried out to retrieve pertinent articles. Studies that meet the inclusion criteria will be eligible for selection. The process of selecting eligible studies will then be summarized on a PRISMA-ScR chart. Collated in data-extraction tables will be authorship information, publication date, methods and findings. The findings, key arguments and themes will be analyzed using a thematic analysis and summarized using a narrative summary.

## Conclusion

This review will contribute to the existing knowledge on parental preferences for vaccine reminder strategies and their usefulness in increasing childhood vaccination rates. The findings will inform and improve public health strategies aimed at boosting vaccine uptake among children.

## 1. Introduction

Vaccination is a critical public health intervention that protects against infectious diseases by stimulating the body's immune system to develop immunity [1]. According to the World Health Organization (WHO) [2], vaccines prevent diseases by enhancing the body's natural defence mechanisms. Childhood immunization plays a crucial role in protecting against at least 15 vaccine-preventable diseases such as measles, tetanus, pneumonia, diarrhea, pertussis, haemophilus influenza type B, hepatitis B, diphtheria, rubella, mumps, varicella, poliomyelitis, meningitis, Influenza, and Hepatitis A [3,4]. Despite the well-documented benefits of vaccination, vaccine-preventable diseases continue to pose significant health threats, particularly among children [5]. For instance, measles remains a leading cause of mortality, with one in every 20 infected children developing severe complications such as pneumonia, which accounts for an estimated 700,000 deaths annually among children under five [5,6].

Global efforts aim to increase vaccination coverage and reduce disease resurgence. The United Nations' Global Immunization Agenda 2030 seeks to avert an estimated 51.5 million deaths, with measles accounting for a significant proportion of these preventable fatalities [7–9]. However, recent trends indicate a decline in childhood vaccination rates in Canada and the United States of America (USA), raising concerns about potential outbreaks [10]. In the USA, the measles-mumps-rubella (MMR) vaccination rate declined from 79% in 2014 to 73% in 2019 and further to 61% in 2021 [10]. During the COVID-19 pandemic, vaccination disparities became evident, with Black children in the USA receiving COVID-19 vaccines at lower rates

(1%) compared to Hispanic (2.5%) and White children (4.5%) [11]. Similarly, in Canada, data on vaccination coverage among Black children is limited, but available evidence suggests disparities exist. For instance, Statistics Canada reported that Black Canadian children had lower vaccination coverage (73%) compared to other racialized children (86%) [12].

One significant factor contributing to suboptimal vaccination rates is parents missing scheduled immunization appointments [13]. The WHO estimated that in 2023 alone, approximately 14.5 million children missed at least one routine vaccination dose, with 33 million children missing measles vaccines in 2022 [14,15]. Missing vaccinations increases the risk of severe health outcomes, including encephalitis and blindness due to measles [14,16]. In Canada, an estimated 300,000 children either missed or were late in receiving routine immunizations, and one in six parents expressed hesitancy toward vaccinating their children [17].

Parental knowledge, awareness, and literacy levels significantly influence childhood vaccination uptake. Hargono et al.'s study [13] showed that parents with insufficient vaccination knowledge are up to six times more likely to miss immunization opportunities than those with greater awareness. Providing parents with timely vaccine reminders can enhance adherence to immunization schedules and improve overall vaccination rates [18,19]. Vaccine reminders encompass digital (e.g., text messages, mobile apps) and non-digital (e.g., mailed notices) strategies that inform parents of upcoming immunization appointments [20].

Another critical issue influencing vaccine uptake is racial and socioeconomic disparities. Research indicates that Black parents in Canada and the USA are more likely to experience barriers to vaccination access, contributing to lower immunization rates among their children [21]. Data from the Government of Canada revealed that Black children had lower coverage for key vaccines, such as rotavirus (69%) and Hib (65%), compared to White children (86% and 77%, respectively) [22]. In the USA, vaccine coverage among Black children was reported at 70.54%, compared to 82.74% among White children [23]. Structural barriers, such as limited healthcare access, lower income levels, and vaccine hesitancy, further exacerbate these disparities [24].

Vaccine reminder strategies have been identified as an effective intervention to increase immunization rates [25]. Studies show that parents prefer receiving reminders from healthcare providers, as forgetfulness is a common reason for missing vaccine appointments [25,26]. Increasing vaccine coverage through effective reminder systems aligns with the United Nations Sustainable Development Goal (SDG) 3.2, which aims to reduce childhood morbidity and mortality [27]. However, with a limited understanding of parents' vaccination reminder interests and preferences, especially the Black parents in Canada and the USA, most effective strategies [28–30] would remain unexplored. Additionally, there is a lack of evidence on which reminder strategies are most effective in improving childhood vaccination rates in diverse populations [29,31].

This scoping review protocol aims to map the existing literature on vaccine reminder strategies available to parents in Canada and the United States, focusing on racial and socioeconomic disparities. Specifically, the review will: a) Identify the extent of research on vaccine reminder strategies and parental preferences; b) Explore gaps in immunization reminders among parents in the literature, especially concerning Black parents in Canada and the USA.

For this study, effective vaccine reminder strategies are defined as interventions that successfully improve childhood vaccination adherence [32]. This review will assess these strategies based on parental preferences and findings from intervention studies [33]. By synthesizing existing evidence, this review will contribute to developing targeted interventions to enhance childhood vaccination rates, particularly among historically underserved populations.

## 2. Methods

In June 2024, the scoping review title was registered with the Open Science Framework (OSF). The proposed scoping review will follow the JBI methodology for scoping reviews and use Peters et al.'s [34] recommendation as the framework guiding the scoping review, as originally developed by Arksey and O'Malley [35]. The six steps include:

1)identifying the research question, 2) aligning the inclusion/exclusion criteria to the review objective, 3) identifying relevant studies (search strategy), 4) extracting the data, 5) collating and summarizing the results, 6) Consultation.

### 2.1. Identifying the review question

This review uses the PCC (population, concept, context) approach as outlined by Pollock et al. [36] to formulate the research question and sub-questions as presented in Fig 1. These will be used to determine the types of childhood immunization reminder strategies available to parents in Canada and the USA.

### 2.2. Inclusion and exclusion criteria

The inclusion and exclusion criteria will be studies (quantitative, qualitative and mixed studies including gray literature) focusing on parents with children six years or younger in Canada and the United States of America and published in the English language. Additionally, grey literature that meets the inclusion and exclusion criteria will be included. These will comprise of theses and dissertations, conference papers and editorials, including published data, will be included to ensure that the data for the review is reliable and extensive. Commentaries will be excluded as they may not provide reliable data. Grey literature that meets the inclusion and exclusion criteria will be included in the review. Quality assessment of the studies, including grey literature, will be manually examined using JBI quality assessment tools to determine studies that are eligible. The summary of these eligibility criteria is in Table 1 In addition, the Participants, Concept and Context (PCC) conception elaborates on the eligibility criteria.

### 2.3. Participants

Studies on immunization reminder strategies in Canada and the USA for parents with children under 6 years old will be included in the review. The rationale is that in Canada and the USA, immunizations are given to children who are up to age 6 years. Children in this study do not include adolescents, and therefore, all studies that involved only adolescents will be excluded. Only studies that report findings on younger children will be retrieved and used for this scoping review. However, eligible studies conducted among adolescents and younger children will be included if they report on younger children below 6 years.

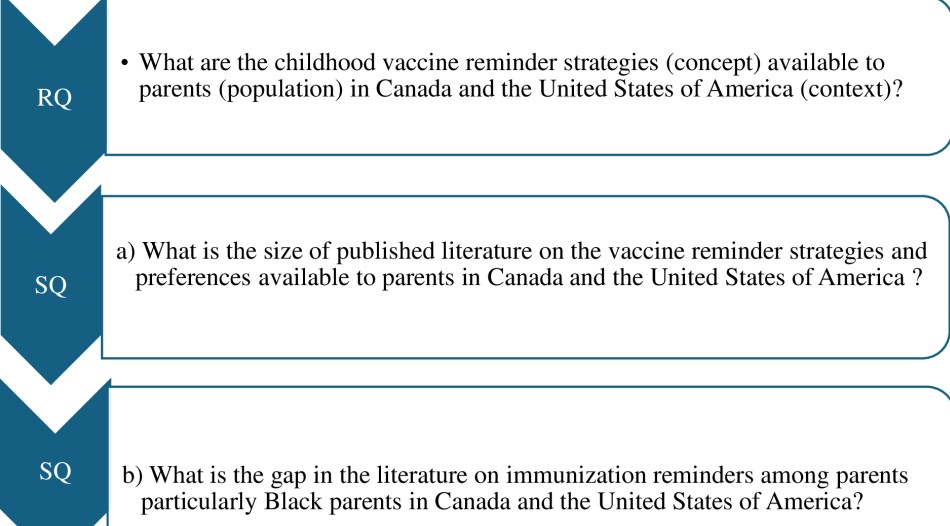

**Fig 1. Research Questions (RQ) and Sub-questions (SQ).**

---

## 2.4. Concept

The main concept of interest for this review is immunization reminder strategies provided to parents to enhance vaccination uptake for their children. Studies focusing on immunization programs or interventions but not immunization reminders will be excluded. This study defines reminders as digital and non-digital communication strategies used to notify parents about vaccination appointments. Digital reminders include text messages, telephone calls and email notifications, while non-digital reminders include appointment endorsements in health cards and family-based reminders.

## 2.5. Context

The search strategy will include limits on setting, language, and publication years. Studies conducted in North America, specifically Canada and the USA, will be eligible for inclusion. Existing literature highlights a lack of research on vaccine reminders for parents, particularly among the Black population, who may have reservations about vaccination due to historical experience, migration, and socio-ecological factors, including racial discrimination experiences [29,31]. Therefore, the selected North American countries (i.e., Canada and the USA) will be included in the review to identify gaps in knowledge about vaccine reminders for parents, especially Black parents.

This scoping review will critically examine full-text articles published in English, as the reviewers do not have expertise in other languages spoken in North America and are constrained by time. Studies published between 2015 and 2025 will be included. The ten-year time frame was chosen to ensure the review reflects current practices, preferences, technological trends, and evolving parenting approaches while considering shifts in healthcare access.

## 2.6. Identifying relevant studies

The second stage of the scoping review will portray the systematic search strategy of databases and journals that were explored including a hand search of journals and gray literature. The search terms of the review are also described in this stage [34,37].

## 2.7. Search strategy

The expertise and partnership of a health science librarian facilitated the search strategy using formulated search terms, keywords and Medical Subject Headings (MeSh). These will be combined using the Boolean Operators (AND, OR and NOT): The search terms that will be used for the review are similar to: immunization, parents, child*, infant*, vaccine reminders, reminder* or prompt* or "text message* reminder*" or text* or email* or call* or "reminder system*" or "digital reminder*" or phone call* or "telephone call*" or "follow up call*" or follow-up or following-up; b) vaccin* or immuni* or inoculat* or (MH Immunization*).mp.; c) parent* or caregiver* or mother* or father* or guardian* or famil* or

**Table 1. Inclusion and exclusion criteria.**

| CRITERIA | INCLUSION | EXCLUSION |
|---|---|---|
| Type of publication | Peer reviewed articles from 2015–2025<br>Gray Literature | Articles published earlier than 2015 |
| Types of research studies | Qualitative, quantitative, mixed methods studies and literature reviews. Randomised and non-controlled trials, observational and Cross-sectional studies. Discussion papers, theses and dissertations, conference papers, book chapters, editorials, book reviews. | Commentaries will be excluded as they may not provide reliable data for the review. |
| Study population | Parents having children under 6 years old | Parents having only adolescent children |
| Publication language | English | Exclude all publications that were not written in English. |
| Settings | Canada and United States of America. | Exclude studies in north America that were not conducted in Canada and the United States of America. |

(MH Parent*).mp. The protocol will involve a preliminary search using the keywords in the search strategy in one database, CINAHL (see S1A Table) for the search terms for the proposed review. In addition, the search strategy will aim to locate both published and unpublished studies (see S1B Table). The search will be carried out in the Cumulative Index to Nursing and Allied Health Literature (CINAHL) [38–40]. Medline (OVID) and EMBASE [38,41]. Besides databases, hand-searching of journals and gray literature will be carried out to retrieve pertinent articles and book chapters.

Gray literature will include theses and dissertations from appropriate databases. Manual and electronic searches from public health organizations' websites such as the Public Health Agency of Canada and the United States Center for Disease Control will be conducted to search for discussion papers, conference papers, commentary, editorials and book reviews. The grey literature that will be included are those that have been published to improve the quality of information for the review.

### 2.7. Study/source of evidence selection

The articles will be selected from qualitative, quantitative and mixed-methods studies. Data will be selected from articles that involved randomized and non-controlled trials, observational and Cross-sectional studies, discussion papers, theses and dissertations, conference papers, book chapters, editorials and book reviews. Findings from both peer-reviewed publications and gray literature will be reviewed. The relevant articles published between 2015 and 2025 will be selected from primary sources to broaden the understanding [34] about vaccine reminders among parents in the selected North American countries. This review will consider the eligibility of, and include, experimental and quasi-experimental studies such as randomized and non-randomized control trials, cohort, longitudinal and observational studies, as well as cross-sectional studies.

Furthermore, qualitative studies, including but not limited to phenomenology, grounded theory, and ethnography, will be included. In addition, opinions and texts that were broadcasted in the media from established agencies, like the Alberta Health Services webpage, will also be added, as these may add valuable childhood immunization information to the review.

Following the search, all identified citations will be downloaded using Research Information Systems (RIS) format, collated and saved into Zotero, a free reference management system. The Systematic Review Accelerator (SRA) software will be used to remove all duplicates. The articles will be imported into Joanna Briggs Institute System for the Unified Management, Assessment and Review (JBI SUMARI) software for screening.

The search results and inclusion process will be presented in a PRISMA-ScR flow chart to identify the number of articles downloaded, screened, duplicated and removed and the records used for the analysis (see S1 Fig).

### 2.8. Data extraction

This protocol will provide a consistent method for the data extraction process to address the question of what childhood immunization reminders are available to parents in Canada and the USA. The data extraction process will involve three stages (see S1C Table)

Two independent reviewers (MA & VP) will screen the titles and abstracts of articles in the first stage following the eligibility criteria and to ensure that there are no biases in the selection process. In the second stage, the full text and citations of potentially relevant sources will be retrieved and imported into the JBI SUMARI software for detailed screening and data extraction. The detailed review of the full-text screening is to get comprehensive information about using vaccine reminders that are usually found in the methods and results sections of the literature. At each phase of the screening, conflicts will be resolved through a dialogue among the reviewers or with an additional reviewer (MA & SM). For instance, if the articles do not fully meet eligibility requirements and the two independent reviewers do not agree to their inclusion, a third additional reviewer (AM) will be invited (by MA) to resolve the conflicts.

Furthermore, the author will employ a data extraction table generated by the JBI SUMARI software and adapted by the review team to extract data for the screening process (see S1D Table). The table will summarize information about authorship, publication information, study purpose, methods (population, concept, context), findings or key arguments and conclusion. Data such as type of reminders, immunization, age of children, and vaccine uptake will be extracted from both published and unpublished studies.

## 3. Data analysis and presentation

The data collated from all relevant and eligible full-text articles will be analyzed using all six iterative processes of Braun and Clarke's [42] qualitative thematic analysis. The processes involve the following iterative steps:

a)reviewing and increasing familiarization with data, b) identifying and deriving codes from data, c) using deductive approaches to clarify codes that align with the review aims and research questions, d) reviewing and conceptualizing the themes, e) defining the themes and f) producing the final report.

Thematic analysis is an iterative process that involves reading and re-reading the literature to understand how the ideas flow logically, sorting out categories, and identifying themes that arise from repetitive ideas and patterns from the data being analyzed [42]. Thematic analysis will be used to identify themes from the extracted data and discussed using a narrative summary.

The results of this protocol will follow the PRISMA-ScR checklist for reporting guidelines (see supporting information S1E Table) [43]. A meta-synthesis will be used to analyze the results of the review [44]. Meta-synthesis will be employed to interpret qualitative data and to provide an understanding of parental reminder strategies that improve childhood vaccinations. Qualitative data analysis, specifically thematic analysis will be employed to interpret and combine both textual and numerical data, which will be summarized using a narrative summary. Although different study designs will be analyzed, the final report will be summarised using a narrative summary [45]. Thus, a narrative summary will be used to interpret both the quantitative and qualitative data during analysis. A table will be used to meaningfully reflect the identified themes from the articles to understand the immunization reminder strategies available to parents in North America. In addition, differences identified between Black and non-Black parents in the uptake of immunization reminders will be presented. It is optional to conduct an appraisal of studies used for scoping reviews. However, in this study, the JBI SUMARI appraisal tool will be used to appraise the full-text articles that will be reviewed.

Furthermore, although the review involved studies published over the past 10 years in North America about vaccination reminder strategies for childhood vaccinations, an initial and cursory search indicates that Canada has limited studies that considered parental reminders for children under the age of six years which could skew the results to only one setting (USA).

### 3.1. Significance of the study

The study findings will provide information on vaccine reminder strategies for parents, including Black parents, and contribute to our understanding of the existing racial disparities in childhood vaccination rates. Thus, it is expected that the findings of this study will provide more knowledge about the vaccine reminder strategies that Black parents and other racialized populations prefer to enhance vaccine uptake, particularly for Black parents who often have low childhood vaccinations. By enhancing our awareness of preferred immunization reminder strategies among parents, the findings can inform public health nursing practice, intervention research targeting childhood vaccinations such as measles, parental education programs, and health policies in North America. Specific recommendations and lay summaries will be made to inform nursing practice, education, research and administration about the preferred reminder strategies for parents, most especially Black parents, for childhood vaccinations, including Black community agencies, respectively. In addition, the findings of the study will be disseminated at relevant conferences using oral and poster presentations and the manuscript

will be published in a peer-reviewed journal. More importantly, gap in the literature on parental vaccine reminder preference in Canada and the USA will be identified and used to inform future research.

## 3.2. Implications

The implication is that future studies in Canada, can consider exploring health systems strategies to improve childhood vaccination among parents, and with an equity-seeking group such as Black parents. Future research can explore cultural differences that influence the preference for reminder strategies and the content of those reminder strategies. In addition, this study will help to provide information for nursing and public health's child immunization policy about the reminder strategies and the complimenting factors such as parental health education which can promote critical vaccine uptake. Thus, this study will inform public health authorities about the specific preferred reminder strategies for parents, especially Black parents in North America, to develop culturally sensitive vaccination programs as well as to improve childhood vaccinations. For instance, creating awareness to the health authorities about what Black parents prefer, e.g., a combination of reminders, such as emails and phone calls, could inform how these strategies are used to involve parents in participating more in childhood vaccinations. In addition, considering that culturally, Black parents tend to emphasize family and community, which aligns with the African philosophy of Ubuntu [46,47], adding this critical stakeholder key in promoting vaccination is long overdue. Similar findings of reminders from family and social contacts or significant guardians in this review will inform public health authorities to utilize culturally appropriate strategies to advance vaccine uptake among parents, particularly Black parents.

## 3.3. Strengths and possible limitations

The study will identify gaps in the literature and highlight racially specific findings by identifying differences in the uptake of immunization reminders between Black parents and non-Black parents. It is expected that including all parents will prevent bias in the findings.

The scoping review protocol may face limitations in capturing all eligible data. This limitation will be addressed by considering four main databases that will be utilized to identify studies conducted in the past ten years, thereby covering most of the published work on the scoping review topic. Another anticipated limitation is that the scoping review will be restricted to studies conducted in English due to time constraints.

## Supporting information

**S1A Table. Literature search terms using CINAHL.**
(DOCX)

**S1B Table. Three-step literature search strategy.**
(DOCX)

**S1 Fig. The figure shows a sample PRISMA chart for screened and included data.**
(TIF)

**S1C Table. Scoping review data extraction stages.**
(DOCX)

**S1D Table. Scoping review protocol data extraction table.**
(DOCX)

**S1E Table. PRISMA-ScR Checklist.**
(PDF)

## Acknowledgments

Profound gratitude goes to Dr Mary Asirifi of the University of McEwan, Canada, who contributed immensely by providing ongoing academic and non-academic support during the completion of this protocol. Her continued motivation and critique allowed MA, the first author, to bounce several ideas for the doctoral coursework, and progress of this work. Sincere appreciation is given to Dr. Frederick Anafi of the Korle Bu Nursing Training College, Ghana for taking on a reader role and proofreading the document during the preliminary stages of the completion of the scoping review study ideas.

## Declarations

The scoping review protocol was written solely under the guidance of my supervisors, whereby all critiques have been duly acknowledged. The principal author and some of the supervisors are Black parents who explored how parents, including an equity-related population of Black parents, received immunization reminders before vaccinating their children.

## Author contributions

**Supervision:** Vivian Puplampu, Sithokozile Maposa, Akram Mahani, Mary Chipanshi.

**Writing – original draft:** Matilda Anim-Larbi.

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
