## [Decision Letter · Decision Letter 0]

6 Jan 2025

Dear Dr. Anim-Larbi,

Thank you for submitting your manuscript to PLOS ONE. After careful consideration, we feel that it has merit but does not fully meet PLOS ONE’s publication criteria as it currently stands. Therefore, we invite you to submit a revised version of the manuscript that addresses the points raised during the review process.

**ACADEMIC EDITOR:  ** Dear Authors, The manuscript needs major revisions. Respond point by point to the requests of the reviewers.

Best regards

We look forward to receiving your revised manuscript.

Kind regards,

Omar Enzo Santangelo

Academic Editor

PLOS ONE

Journal Requirements:

2. Thank you for stating the following in your Competing Interests section:  The authors of this scoping review would like to declare that there are no conflicts of interests, monetary, personal, academic or professional bias associated with this project. The Joanna Briggs Institute of Evidence Synthesis is a reference for further guidance pertaining to this work. 

Reviewers' comments:

Reviewer's Responses to Questions

**Comments to the Author**

1. Does the manuscript provide a valid rationale for the proposed study, with clearly identified and justified research questions?

Reviewer #1: No

Reviewer #2: Yes

Reviewer #3: Yes

Reviewer #4: Yes

Reviewer #5: Yes

2. Is the protocol technically sound and planned in a manner that will lead to a meaningful outcome and allow testing the stated hypotheses?

Reviewer #1: Partly

Reviewer #2: Yes

Reviewer #3: Yes

Reviewer #4: Yes

Reviewer #5: Yes

3. Is the methodology feasible and described in sufficient detail to allow the work to be replicable?

Reviewer #1: No

Reviewer #2: Yes

Reviewer #3: Yes

Reviewer #4: Yes

Reviewer #5: Yes

4. Have the authors described where all data underlying the findings will be made available when the study is complete?

Reviewer #1: Yes

Reviewer #2: Yes

Reviewer #3: Yes

Reviewer #4: No

Reviewer #5: Yes

5. Is the manuscript presented in an intelligible fashion and written in standard English?

*PLOS ONE*

Reviewer #1: Yes

Reviewer #2: Yes

Reviewer #3: Yes

Reviewer #4: Yes

Reviewer #5: Yes

You may also provide optional suggestions and comments to authors that they might find helpful in planning their study.

Reviewer #1: This paper describes a scoping review to map the literature on immunization reminder strategies for parents. As we are in a time of declining immunizations, this work is very important and could potentially be very impactful. However there is a lack of clarity around the specific rationale/focus for this study, and it is unclear why the authors are embarking on a scoping review and not a systematic review for this work, particularly as many of the methods pertain to a systematic review protocol.

Specific comments:

The abstract includes the phrase, “with a focus on supporting Black parents”, but what does this specifically mean? Does this mean immunization reminders are needed for Black parents in particular due to vaccination rates, or is it that reminders have been found to be particularly useful to this group of parents? Additional information in the background/rationale of the paper to justify this focus would be helpful.

The introduction emphasizes the importance of vaccines, but the first paragraph seemed to be about the importance of strategies to improve the vaccines themselves which seemed to differ from the purpose of the article? It would perhaps be useful to begin with information about vaccination and then discuss vaccine uptake and whether there are differences among different patient populations. It would also be helpful to provide additional context on the trends in vaccination rates over time.

Importantly, (linking to the comment above about the abstract) the general rationale for focusing on vaccine reminders wasn’t clear. There were arguments posed about lack of knowledge about vaccines, and about disparities between parents belonging to different ethnic groups, but the arguments and references provided did not clearly show how providing reminders would resolve these issues.

The focus on North America was also not made clear – why only North America, particularly when some of the other literature cited in the introduction focused on other countries?

The objective referred to examining the effectiveness of vaccine reminder strategies – it might be helpful to include the word ‘effectiveness’ in the title to make it clearer.

Towards the end of the introduction there is some detail that may be best allocated to the methods section (e.g. beginning on page 6, line 121).

I would suggest moving the ‘significance of the study’ section towards the end of the manuscript (unless this follows a prescribed PLOS One formatting preference that I missed). It also was not clear as to how this study will contribute to understanding of racial disparities in vaccination rates – if this will come out of the results from this study perhaps it will become clearer by moving this section to the end of the protocol paper. The statement about identifying areas for future research I think is making an assumption of what you will find in the results of the scoping review and I would suggest removing it.

I would also recommend removing the section on ‘identifying the research question’. Figure 1 could be included but might fit better elsewhere. It would also help to specifically align the various RQ’s in Figure 1 with the objectives of this study – some of the RQ listed in figure 1 seemed to extend beyond the plans referred to at the end of the introduction.

Methods:

Please provide the rationale for focusing on 2004-2024. An explanation was provided in the methods for why children up to age 6 years were included, but rationale for North American studies only was not provided (which could be in the introduction, as suggested above).

The search strategy may find a lot of articles intended for reminders to adults – I wonder if it would be helpful to consider including a term for children and infants? How will the grey literature be searched/used?

In general throughout the methods there seemed to be some confusion over the process for a scoping review vs a systematic review. Both ‘charting the data’ and ‘data extraction’ were referred to – which one will be done? The terminology throughout could be made consistent to fit with the specific study type. Related to this, I was not clear why the authors were using the PRISMA checklist and not PRISMA-ScR if it’s a scoping review – although both appeared in the appendix?

The strengths and limitations section is quite brief and could be expanded.

In terms of implications, how do the authors think the findings may inform and improve public health strategies to boost vaccine uptake?

Many thanks for the opportunity to review this work and I hope these comments are helpful to the author, particularly as they develop their personal PhD project.

Reviewer #2: The introduction includes some redundant statements, particularly about the benefits of immunization.

Using approaches and strategies interchangeably might not right as the two are different

Certain claims, such as the effectiveness of reminders, lack specific evidence or references in the introduction section.

The manuscript does not explicitly mention how findings will be synthesized to inform practical applications or policy recommendations.

The discussion lacks depth in addressing the challenges of synthesizing diverse data sources (e.g., qualitative vs. quantitative).

The implications for Black parents are highlighted but not explored in detail in the dicscussion.

Reviewer #3: The protocol shows promise. However, it suffers from methodological gaps, unclear objectives, and limited justification for some decisions. Addressing these issues comprehensively will significantly improve the manuscript's quality and relevance.

Specifically,

1. The inclusion of studies spanning 20 years may no longer be fully relevant given the rapid advancements in technology, particularly in communication and healthcare systems. Over this period, significant changes have occurred, including widespread access to free information and the growing use of digital tools such as smartphones and automated reminders. It is recommended to limit the inclusion period to more recent years to reflect on the current practices, preferences, and technological trends while addressing generational shifts in parenting and access to healthcare resources.

2. There appears to be a typographical error under the "Settings" section S1A Table, where the inclusion and exclusion criteria mentioned both Canada and US.

3. While the title states that the study focuses on "North America," the inclusion criteria are limited to Canada and the United States. The scope should be expanded to genuinely include all North American countries for consistency with the title.

4. The title of the manuscript is vague and does not explicitly communicate the central focus on Black parents, despite significant emphasis on this demographic in the text.

5. It is unusual and generally unnecessary for a protocol intended for publication to explicitly mention that it is part of graduate work. Publications are expected to focus on the research content itself and its contribution to the field rather than on the context of how or why the study was developed. Please revisit some sections.

Reviewer #4: Rationale and Research Questions:

The manuscript provides a compelling rationale for the proposed scoping review by emphasizing the public health significance of childhood immunization and the resurgence of vaccine-preventable diseases in North America. The research question is well-structured, clearly identifying the focus on parental immunization reminder strategies. However, greater specificity about the unique contributions of this review compared to existing literature could enhance the manuscript's impact. For instance, while the racial disparities in vaccination uptake are discussed, the protocol could expand on how it plans to address the gap in understanding the cultural or systemic barriers specific to Black parents.

Methodological Framework:

The manuscript demonstrates a rigorous approach, adopting the JBI methodology and building upon the framework by Arksey and O’Malley. However, it would benefit from further details about how the inclusion and exclusion criteria will be operationalized in ambiguous cases (e.g., studies addressing both adolescents and younger children). Additionally, the authors might consider discussing the potential challenges of using gray literature and how they will ensure its quality.

Data Availability and Reproducibility:

While the protocol mentions using a systematic approach for data extraction and review, it does not explicitly state where the data underlying the review's findings will be made publicly available after completion. Including this information would align with best practices in transparency and reproducibility.

Parental Reminder Strategy Definitions:

The manuscript defines reminders broadly as digital and non-digital strategies. While this inclusivity is commendable, providing examples or a taxonomy of reminder strategies in the methodology section would help clarify how the study will categorize diverse strategies encountered in the literature.

Significance and Implications:

The study's significance in addressing racial disparities is important, but the manuscript could better articulate how its findings might be operationalized into practical interventions or policies. For example, it could outline how public health authorities could use these insights to design culturally sensitive vaccination programs.

Writing and Presentation:

Overall, the manuscript is well-written and intelligible. However, certain sections could benefit from tighter editing to improve clarity. For instance, the repeated references to studies with similar findings might be synthesized for conciseness. Additionally, using consistent terminology throughout the manuscript (e.g., interchangeably using "reminder strategies" and "reminder preferences") would reduce potential confusion.

Engagement of Black Parents:

Given the focus on racial disparities, the authors should consider elaborating on how they will account for potential biases in the included studies that could affect the representation of Black parents. Explicitly stating a plan to address or discuss these biases in the analysis would strengthen the study's conclusions.

Figures and Tables:

While the manuscript includes references to figures and tables, their clarity and relevance are not fully detailed in the narrative. It would be beneficial to ensure all supplementary materials are cohesive and directly support the primary text.

Recommendations:

Clearly state where and how the final dataset will be made accessible.

Elaborate on potential cultural or systemic barriers affecting Black parents' engagement with immunization reminder systems.

Provide a more detailed breakdown of the types of reminder strategies and how they will be categorized during analysis.

Address how quality will be assessed for gray literature and other non-peer-reviewed sources.

Ensure consistency in terminology and provide clearer links between the research questions, methodology, and anticipated outcomes.

Section-Specific Areas of improvement:

Rationale and Research Questions

The introduction does not clearly specify the unique contribution of this review to the existing body of literature. While it mentions disparities among Black parents in North America, it lacks detail on the specific knowledge gaps this study aims to address.

Methodological Framework

The inclusion and exclusion criteria are broadly stated without providing detailed handling of ambiguous cases, such as studies involving mixed age groups or interventions not exclusively targeting parents.

Data Availability

The manuscript mentions data extraction and analysis but does not address plans for data sharing or transparency in making extracted data available for reproducibility.

Parental Reminder Strategy Definitions

The types of reminder strategies included are vaguely described as "digital and non-digital methods" without offering concrete examples or a taxonomy to guide the reader.

Significance and Implications

The discussion of implications is generic, stating that findings will address disparities in vaccination uptake, but it does not specify how these findings might inform practical or policy-level interventions.

Writing and Presentation

The manuscript contains repetitive references to the effectiveness of digital reminders without synthesizing or streamlining these findings to improve readability and flow.

Engagement of Black Parents

The manuscript does not evaluate or discuss potential biases in the included studies concerning the representation of Black parents or how these biases might impact findings.

Figures and Tables

The figure summarizing the search strategy lacks sufficient detail to allow for an independent replication of the methodology, such as the specific databases searched and the keywords used.

Reviewer #5: Authors have taken only 20 years data. Is it possible to take 50 years of data? Still the article looks good for publication.

**Do you want your identity to be public for this peer review?** For information about this choice, including consent withdrawal, please see our Privacy Policy

Reviewer #1: No

Reviewer #2: **Yes: ** Gyesi Razak Issahaku

Reviewer #3: No

Reviewer #4: No

Reviewer #5: **Yes: ** Ripan Biswas

---

## [Author Response · Author response to Decision Letter 1]

6 Mar 2025

We would like to thank the reviewers for the comments. The responses to reviewers have been tabulated and attached to the submission. I

---

## [Decision Letter · Decision Letter 1]

4 Apr 2025

A Scoping Review Protocol on Childhood Immunization Reminder Strategies Available to Parents in Canada and the United States of America

PONE-D-24-45802R1

Dear Dr. Anim-Larbi,

We’re pleased to inform you that your manuscript has been judged scientifically suitable for publication and will be formally accepted for publication once it meets all outstanding technical requirements.

Kind regards,

Omar Enzo Santangelo

Academic Editor

PLOS ONE

Additional Editor Comments (optional):

Reviewers' comments:

Reviewer's Responses to Questions

**Comments to the Author**

1. Does the manuscript provide a valid rationale for the proposed study, with clearly identified and justified research questions?

Reviewer #3: Yes

Reviewer #5: Yes

2. Is the protocol technically sound and planned in a manner that will lead to a meaningful outcome and allow testing the stated hypotheses?

Reviewer #3: Yes

Reviewer #5: Yes

3. Is the methodology feasible and described in sufficient detail to allow the work to be replicable?

Reviewer #3: Yes

Reviewer #5: Yes

4. Have the authors described where all data underlying the findings will be made available when the study is complete?

Reviewer #3: Yes

Reviewer #5: Yes

5. Is the manuscript presented in an intelligible fashion and written in standard English?

*PLOS ONE*

Reviewer #3: Yes

Reviewer #5: Yes

You may also provide optional suggestions and comments to authors that they might find helpful in planning their study.

Reviewer #3: The authors have addressed my specific suggestions, and the manuscript has improved in clarity and structure.

Reviewer #5: The manuscript presents a well-structured scoping review protocol that addresses an important public health topic—childhood immunization reminder strategies for parents in Canada and the USA. The study is timely, given the global emphasis on improving vaccination coverage and addressing vaccine hesitancy. The protocol adheres to established methodological frameworks (e.g., PRISMA-ScR, Arksey & O’Malley) and demonstrates rigor in its design. However, minor revisions are recommended to enhance clarity, specificity, and methodological transparency.

**Do you want your identity to be public for this peer review?** For information about this choice, including consent withdrawal, please see our Privacy Policy

Reviewer #3: No

Reviewer #5: **Yes: ** Ripan Biswas

---

## [Editor Report · Acceptance letter]

PONE-D-24-45802R1

PLOS ONE

Dear Dr. Anim-Larbi,

I'm pleased to inform you that your manuscript has been deemed suitable for publication in PLOS ONE. Congratulations! Your manuscript is now being handed over to our production team.

Kind regards,

on behalf of

Dr. Omar Enzo Santangelo

Academic Editor

PLOS ONE